# Spatial Variability in a Symbiont-Diverse Marine Host and the Use of Observational Data to Assess Ecological Interactions

Edwin Cruz-Rivera [1,*], Mohy-El-Din Sherif [2], Salma El-Sahhar [3] and Thomas Lombardi [4]

1 Bioenvironmental Science Program, Department of Biology, Morgan State University, 1700 E. Cold Spring Lane, Baltimore, MD 21251, USA
2 AB Vista NIR Services, AB Agri, Paulerspury N12 7LS, UK; mohyeldin.sherif@abagri.org
3 School of Life Sciences, University of Essex, Colchester CO4 3SQ, UK; salma.elsahhar@essex.ac.uk
4 Department of Information Systems and Technology, University of the Virgin Islands, #2 John Brewers Bay, St. Thomas, VI 00802, USA; thomas.lombardi@uvi.edu
* Correspondence: edwin.cruz-rivera@morgan.edu

**Abstract:** Despite a rich taxonomic literature on the symbionts of ascidians, the nature of these symbioses remains poorly understood. In the Egyptian Red Sea, the solitary ascidian *Phallusia nigra* hosted a symbiotic amphipod and four copepod species, with densities as high as 68 mixed symbionts per host. Correlation analyses suggested no competition or antagonism between symbionts. Ascidian mass, ash-free dry mass per wet mass (AFDM/WM), and both symbiont density and diversity per host, differed significantly among three reefs from El Gouna, Egypt. However, there was no correlation between amphipod, total copepod, or total symbiont densities and host mass or AFDM/WM. A host condition index based on body to tunic mass ratio was significantly related to symbiont density overall, but this positive pattern was only strong at a single site studied. Despite assumptions based on the habit of some of the symbiont groups, our analyses detected little effect of symbionts on host health, suggesting a commensal relationship.

**Keywords:** ascidian; *Bonnierilla*; *Doropygus*; *Janstockia*; *Leucothoe*; *Styelicola*; Notodelphyidae; *Phallusia nigra*; Red Sea

## 1. Introduction

Many marine symbioses are poorly understood and have been often classified based on the taxonomy of the animals involved rather than on quantification of costs and benefits [1–3]. While taxon-based inferences have been informative and often correct, they can obscure fundamental differences in the nature of interactions within a clade and the context dependence of symbioses within a parasitism to mutualism continuum [4–8]. For example, apicomplexan protozoa, which have been largely treated as parasites/pathogens, have been increasingly reported as commensals and mutualists of marine invertebrates and vertebrates [9–12]. A recent article on the purported symbiont diversity of the snail *Littorina littorea* (Linnaeus, 1758) also highlights the perils of assuming symbiont roles without considering alternative hypotheses and the complexity of natural interactions [13]. In that study, a more rigorous sampling within a community context elucidated that previously classified snail endosymbionts were, in fact, transient associates trapped in the mucus matrix secreted by the snail.

A cost–benefit analysis of a pairwise interaction within a community context can elucidate the outcome (and ecological classification) of the association between symbiont and host. A manipulative approach in which hosts and symbionts are grown independently from one another, and together, could offer an ideal method to quantify fitness effects for each interacting species. However, this is not feasible in most cases of obligate symbioses and is difficult to achieve when life cycles require multiple hosts or when endosymbiont presence cannot be confirmed without sacrificing the host.

An alternative approach is to take advantage of a natural experiment: sampling of host populations, and comparing fitness of hosts that harbor differing numbers and kinds of symbionts. While such a method only provides information on the host, it can reveal important density-dependent effects as symbiont loads can change in time and space and the nature of the symbiosis can change along a gradient [7,8,14,15]. Sampling natural symbiont populations can also help unravel interactions between multiple symbionts that inhabit the same host [16]. An important consideration in applying this framework is determining what components of host health or fitness can be measured. Techniques such as calculating gonadosomatic or condition indices have been fundamental in assessing how environmental variables affect allometric relations, life history, structural traits, basic health, and fitness components in aquatic and terrestrial animals [17–26]. In shellfish aquaculture, for example, body to shell mass ratios are widely used proxies of animal health [18,19]. Similar approaches have been applied to more ecological studies of echinoderms [21,22], gastropods [27], bivalves [28], and tube-dwelling polychaetes [29], among others.

Here we use two body condition proxies to evaluate the effects of symbiotic crustaceans on the sea squirt *Phallusia nigra* Savigny, 1816 (Tunicata: Ascidiacea). While believed to be a Red Sea endemic by some, this solitary ascidian has a worldwide distribution and serves as host to several invertebrate symbionts [30,31]. It is a shallow-water species found on hard natural and artificial bottoms at depths of up to 14 m [32–34]. Adults are 4–12 cm long and reproduce throughout the year [32,35]. This ascidian is a common member of fouling communities around the world and recruits year-round, although it is more common at early successional community stages [32,36,37]. Population densities can fluctuate seasonally by an order of magnitude [32,38] and surpass $100/m^2$ within native and invaded ranges, with the highest recruitment densities recorded reaching $> 500/m^2$ in the Red Sea [38–40].

The tunic of *P. nigra* accumulates vanadium, acid, and other secondary metabolites, which serve as chemical defenses against predators and fouling organisms and have been proposed as mechanisms promoting the longevity of adults past the initial recruitment stages [41–43]. A diverse symbiont community has evolved to utilize this defended ascidian host around the world (Table 1). In the Red Sea alone, *P. nigra* hosts the amphipod *Leucothoe furina* (Savigny, 1816), a polychaete worm, and at least seven species of copepods that live within different parts of its body [44–48] (Table 1). Outside of this geographic area, five other amphipod symbionts have been reported from *P. nigra* in Belize, Cuba, Florida, Panama, Brazil and Venezuela (Table 1), although records of *L. spinicarpa* and *L. wuriti* from Brazil have been questioned [49,50]. A pinnotherid crab also inhabits *P. nigra* in the Caribbean [51]. Historically, ascidian amphipod symbionts have been considered commensals [45,52,53], whereas copepods have been classified as both commensals and parasites [54–56]. Rarely have these classifications been related to any host traits [14]. Here, we relate amphipod and copepod densities to three ascidian variables that can help elucidate the nature of the symbioses by detecting potential costs for the host. We also assess possible interactions between symbionts. By comparing animals from three reefs in the Red Sea, we evaluate the role of spatial variation in symbiont–host interactions.

**Table 1.** Listing of all symbionts reported from the ascidian *Phallusia nigra* around the world.

| Symbiont | Geographic Location | References |
|---|---|---|
| Crustacea | | |
| Amphipoda | | |
| *Amphilochus ascidicola* Ortiz and Atienza, 2001 | Caribbean (Venezuela) | [57] |
| *Leucothoe angraensis* Senna, Andrade, Ramos & Skinner, 2021 | South Atlantic (Brazil) | [50] |
| *L. flammosa* Thomas and Klebba 2007 | Caribbean (Cuba) | [57] |
| *L. furina* (Savigny, 1816) | Red Sea (Egypt) | [46] |
| *L. spinicarpa* (Abildgaard, 1789) | North Atlantic (USA) | [58] |

**Table 1.** *Cont.*

| Symbiont | Geographic Location | References |
|---|---|---|
| *L. wuriti* Thomas and Klebba 2007 | North Atlantic (USA), Caribbean (Belize, Panama) | [49,52] |
| Brachiura | | |
| *Tunicotheres moseri* (Rathbun, 1918) | Caribbean (Jamaica, Venezuela) | [35,51] |
| Copepoda | | |
| *Bonnierilla projecta* Stock, 1967 | Red Sea (Egypt, Erithrea) | [44,46] |
| *Doropygus humilis* [1] Stock, 1967 | Red Sea (Egypt, Erithrea) | [44,46] |
| *Janhius brevis* [2] (Stock, 1967) | Red Sea (Erithrea) | [44] |
| *Janstockia phallusiella* Boxshall & Marchenkov, 2005 | Red Sea (Egypt) | [46,59] |
| *Lonchidiopsis tripes* Stock, 1967 | Red Sea (Erithrea) | [44] |
| *Notodelphys ciliata* Schellenberg, 1922 | Red Sea (Egypt) | [60] |
| *Notodelphys steinitzi* Stock, 1967 | Red Sea (Erithrea) | [44] |
| *Paranotodelphys phallusiae* (Gurney, 1927) | Red Sea (Egypt) | [61] |
| *Styelicola omphalus* Kim I.H., Cruz-Rivera, Sherif & El-Sahhar, 2016 | Red Sea (Egypt) | [46] |
| Annelida | | |
| Polychaeta | | |
| *Proceraea exoryxae* Martin, Nygren & Cruz-Rivera, 2017 | Red Sea (Egypt) | [47] |

[1] As *D. apicatus* in [44]; [2] As *Prophioseides brevis* in [44].

## 2. Materials and Methods

*Phallusia nigra* were collected from each of three sites (*n* = 50) around El Gouna, on the Red Sea coast of Egypt (27°23′50.4″ N, 33°40′30.2″ E; Figure 1). Using SCUBA, animals were carefully detached from the substrate and placed individually in resealable plastic bags for transport to the John D. Gerhart Field Station (American University in Cairo, formerly). All organisms were collected with permission from the private administrators of Abu Tig Marina, Mövenpick Hotel, and Zeytouna Beach, as well as the El Gouna local authorities through the American University in Cairo. All specimens came from public areas. Only animals that could be retrieved intact were used in the study. Ascidians were collected randomly at 2–5 m depths along 30 m stretches from reefs around Abu Tig Marina (27°24′34.8″ N 33°40′55.1″ E), Mövenpick Hotel (27°23′41.6″ N 33°41′31.1″ E), and Zeytouna Beach (27°24′06.4″ N 33°41′09.8″ E). The areas of collection were approximately 850 m apart between reefs. All collections were performed over the same ten-day period in October to minimize temporal effects on faunal abundances. El-Gouna is one of the main beach tourism destinations in Egypt and the coastline has been modified by extensive dredging and construction over several decades [62–65]. Nearshore communities have been further affected by sewage and garden runoff and by activities from a local desalination plant [66]. As a result, most local reefs have now low coral, and high algal, cover. Despite being relatively close (ca. 850–900 m from one another), the three reefs sampled had noticeable differences in environmental quality. The reef closest to the Abu Tig Marina lies right off the mouth of the main channel where most charter and commercial boats transit in and out EL Gouna. Suspended sediments were consistently higher, and visibility was considerably lower, in this reef compared to Zeytouna and Mövenpick. Zeytouna had a higher amount of live coral and invertebrate diversity. This is an area frequented by divers and snorkelers and is managed by a private company that enforces fishing and collection restrictions. Mövenpick is southeast of Zeytouna and has a very shallow broad lagoon. Tourists are not discouraged from walking across the patch reefs and reef flat, where signs of trampling are common. However, the slope of the reef breaks several meters deeper than in the other two reefs and is less frequented by divers. While a few studies of environmental impacts for this area are available (e.g., [65,66]), they treat EL Gouna as a single region and, therefore, our description of single reefs is based on qualitative observations over three years of collecting at these sites.

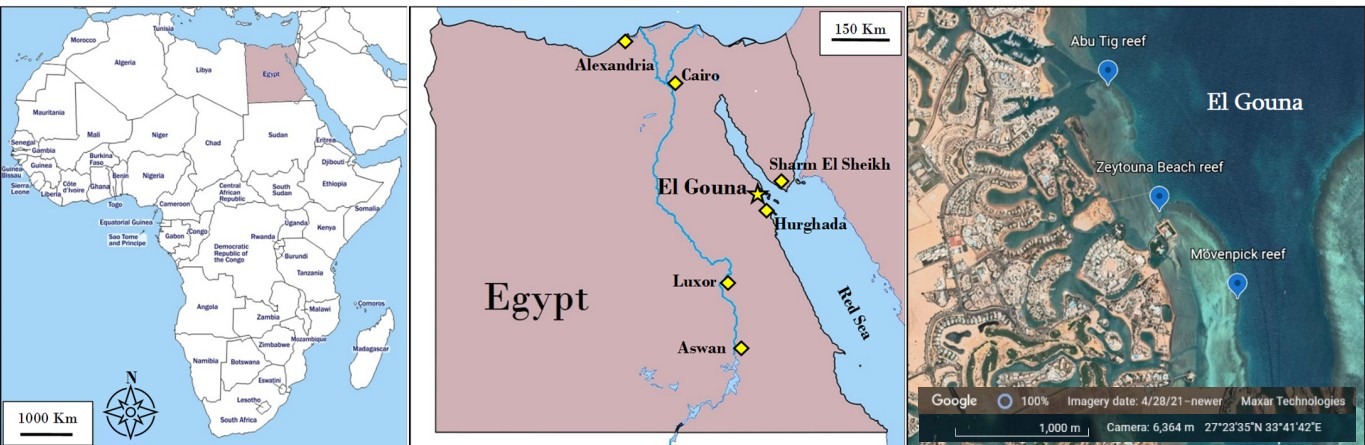

**Figure 1.** Location of the three reefs in El Gouna, on the Egyptian Red Sea coast, where collections took place. Maps adapted from d-maps.com (https://d-maps.com/carte.php?num_car=4338&lang=en and https://d-maps.com/carte.php?num_car=916&lang=en) and Google Earth (https://earth.google.com/web/).

In the lab, ascidians were dissected by making a peripheral incision and separating each *P. nigra* into two halves [46] (Figure 2). The body of ascidians is encased in a protective outer organic layer called the tunic. This is a carbohydrate-based pliable exoskeleton secreted by the epidermis and may incorporate sand, algae, or spicules produced by the animal, depending on the species [67]. Most of the internal cavity is covered by a large modified ciliated pharynx (the pharyngeal basket) that allows the animal to filter feed by inhaling water through a branchial siphon, trapping edible particles, and expelling the water out an atrial siphon. The rest of the organs occupy a visceral cavity, with the genital ducts and anus opening to the atrium. In *P. nigra*, the tunic is smooth, and it readily separates from the body. The visceral mass (digestive and reproductive systems) and pharyngeal basket were carefully inspected because the location of ascidian faunal associates varied within the host according to symbiont species [46,47] (Figure 2). Using the number of associated animals per ascidian, symbiont diversity was quantified by calculating the Shannon-Wiener and Simpson indices for each collected host containing at least one associated species. The total wet mass of each *P. nigra* was used to approximate host size and was calculated by adding the wet masses of the visceral mass and pharyngeal basket with that of the tunic, after gently padding each with absorbent paper to reduce weighing errors due to water content.

To assess host state in relation to symbiont load, two measurements were used. First, percent of ash-free dry mass per wet mass (AFDM/WM) was calculated by drying each dissected *P. nigra* (tunic + body) at 65 °C for three days and then burning in a furnace at 450 °C for eight hours. This measurement of total organic content has been often used as an indicator of nutritional value of plant and algal food to herbivores [68–70], but can also approximate imbalances between the organic and inorganic components of an animal [71,72]. Second, a condition index was calculated as the percent of body to tunic (=[(WM of viscera + pharyngeal basket)/WM of tunic] × 100). The tunic is a thick external protective and supportive organic layer secreted by the epidermis (mantle) of the ascidian body wall. Despite being seldom calcified with spicules and containing some blood vessels, the tunic has many parallels in function and origin with a molluscan shell (an organic matrix as well, but with higher calcification). Thus, our approach is similar to the broadly used meat-to-shell ratio that is applied to approximate health and quality of shellfish in aquaculture and for human consumption [18,73,74].

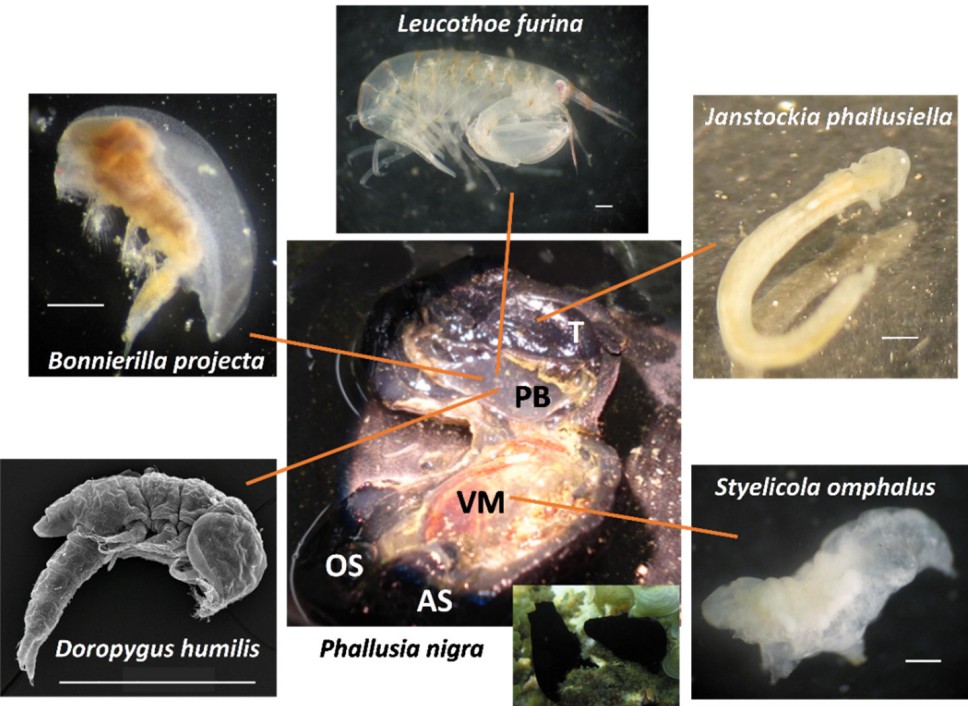

**Figure 2.** Symbionts found in the ascidian *Phallusia nigra* from the Egyptian Red Sea coast. Lines point the typical location of the symbiont inside the host. At the center, a dissected *P. nigra* (length 4.5 cm) is shown with part of the pharyngeal basket removed to expose the inside of the tunic: OS = oral siphon, AS = atrial siphon, PB = pharyngeal basket, T = tunic, VM = visceral mass. Two living specimens of the ascidian are shown in the insert. The scale bars for all symbionts are 0.5 mm. All photos by E. Cruz-Rivera, except *D. humilis* (by Kolbasov, G.A.).

To analyze differences among reefs in host size, condition, and symbiont loads and diversity, we used one-way ANOVA after testing for normality (Kolmogorov–Smirnov tests) and variance homogeneity (Levene's tests). In some instances, departures from these requirements were corrected by log transformation. When significant differences were found, Tukey's HSD tests were used for post hoc comparisons. The non-parametric Kruskal-Wallis test was applied when data did not conform to ANOVA assumptions despite multiple transformations. When significant differences were found, Kruskal-Wallis were followed by Mann-Whitney U tests, adjusted with Bonferroni corrections, for pairwise comparisons. Data for individual symbionts (1) could not be assumed as independent because multiple species could inhabit the same host replicate, (2) were not normally distributed, and (3) included instances where a particular symbiont was absent from a sampled site. These conditions constrained analyses using multifactorial tests (e.g., two-way ANOVA, Scheirer-Ray-Hare test). Instead, we used the non-parametric Kruskal-Wallis test to assess differences among reefs for each symbiont species quantified.

Pearson correlations were used to evaluate potential interactions between *P. nigra* symbionts by comparing densities between species overall and within each sampled reef separately. While understanding the mechanisms of competition between symbionts requires a manipulative approach, correlations and regression analyses can provide useful insights into interspecific relations between symbionts for a given host (e.g., [16]). Linear regressions were used to determine the effects of host size (mass) on symbiont load and the potential effects of symbiont densities on AFDM/WM and condition index, overall and per reef. Log transformations of data were used in various of the analyses above to conform with the assumptions of these parametric approaches. As different symbionts of *P. nigra* inhabit different parts of the ascidian, regressions with AFDM/WM were conducted on whole animals, ascidian tunic, and ascidian body separately, and against total amphipods

(broadly considered commensals), total copepods (often considered parasites for the two families encountered here), and total symbionts.

## 3. Results

One sample from Zeytouna Beach was lost during processing and all analyses herein are based on a sample size of 49 for that site. The amphipod *Leucothoe furina* (Savigny, 1816), the ascidicolid copepod *Styelicola omphalus* Kim I.H., Cruz-Rivera, Sherif & El-Sahhar, 2016, and the notodelphyid copepods *Bonnierilla projecta* Stock, 1967, *Doropygus humilis* Stock, 1967, and *Janstockia phallusiella* Boxshall & Marchenkov, 2005, were all found in *P. nigra* from our collections. However, there were spatial differences in distribution. For example, *D. humilis* were never found in Mövenpick reef ascidians.

Data showed no indication of antagonism or tradeoff in the distributions of these symbionts (Table 2). In contrast, a weak, but significant positive correlation between the number of amphipods and the density of the copepod *B. projecta* was observed when all reefs were analyzed together ($p = 0.037$, Pearson correlation coefficient = 0.171). When the three sites were compared, this correlation was only detected for Mövenpick reef ($p = 0.048$, Pearson correlation coefficient = 0.281). The only other significant correlation found was between the presence of *B. projecta* and the copepod *D. humilis* at Zeytouna Beach ($p < 0.001$, Pearson correlation coefficient = 0.836).

**Table 2.** Correlations between symbiont abundances as proxies for pairwise interactions within ascidian hosts. Analyses were conducted for all studied reefs together and individually. Numbers are *p*-values from two-tailed Pearson correlations. No *Doropygus humilis* were found at Mövenpick reef. Numbers in bold indicate significant correlations. Only positive correlations were detected.

| **All Field Sites** | | *Bonnierilla* | *Doropygus* | *Janstockia* | *Styelicola* |
|---|---|---|---|---|---|
| | *Leucothoe* | **0.037** | 0.251 | 0.502 | 0.773 |
| | *Bonnierilla* | | 0.092 | 0.474 | 0.817 |
| | *Doropygus* | | | 0.579 | 0.078 |
| | *Janstockia* | | | | 0.761 |
| Individual sites | | | | | |
| | *Leucothoe* | 0.593 | 0.526 | 0.863 | 0.360 |
| Abu Tig | *Bonnierilla* | | 0.861 | 0.418 | 0.548 |
| | *Doropygus* | | | 0.641 | 0.774 |
| | *Janstockia* | | | | 0.553 |
| | *Leucothoe* | **0.048** | - | 0.553 | 0.832 |
| Mövenpick | *Bonnierilla* | | - | 0.985 | 0.847 |
| | *Doropygus* | | | - | - |
| | *Janstockia* | | | | 0.731 |
| | *Leucothoe* | 0.645 | 0.421 | 0.81 | 0.657 |
| Zeytouna Beach | *Bonnierilla* | | **<0.001** | 0.950 | 0.741 |
| | *Doropygus* | | | 0.656 | 0.839 |
| | *Janstockia* | | | | 0.755 |

*Phallusia nigra* mean wet mass, a proxy for size, was significantly different among reefs ($p = 0.004$, one-way ANOVA [log-transformed data]; Figure 3). Ascidians were significantly larger at Abu Tig than at Zeytouna Beach. Mövenpick ascidians were intermediate in mass and statistically similar to those in the other two reefs. In contrast, total AFDM/WM of the ascidian hosts was significantly lower at Abu Tig than at Zeytouna Beach, although this difference was <7% ($p = 0.003$, Kruskal-Wallis; Figure 3). AFDM/WM of ascidians from Mövenpick reef was statistically equivalent to that of the other sites. Despite differences in other parameters, condition indices were very similar across reefs ($p = 0.767$, one-way ANOVA; Figure 3).

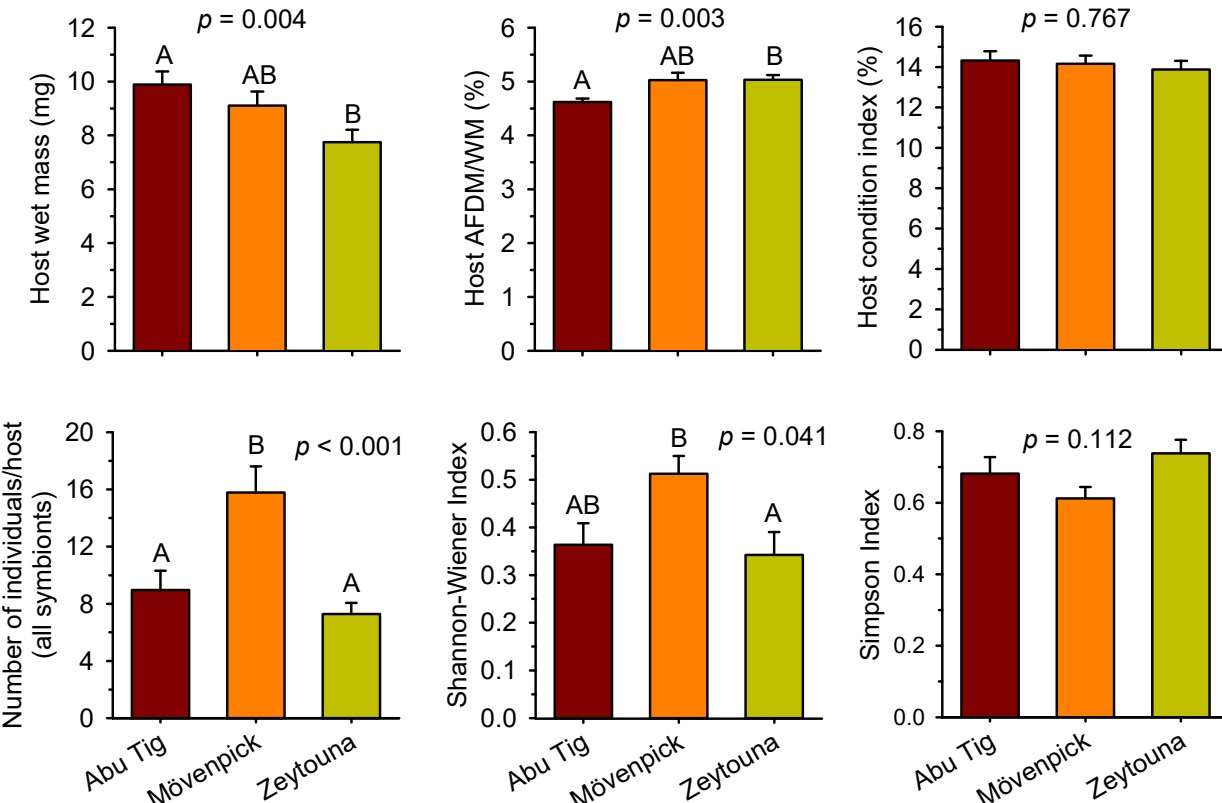

**Figure 3.** Wet mass, ash-free dry mass per wet mass (AFDM/WM), and condition of the ascidian *Phallusia nigra* collected at three reefs from El Gouna, Egypt (top row). Associated fauna (total ascidian symbionts) and two indices of species diversity of symbionts (Shannon-Wiener and Simpson indices) are presented on the bottom row. Bars represent means ± 1 SE. P values are from one-way ANOVA or Kruskal-Wallis tests, followed by appropriate pairwise comparisons as needed (see Methods). Same letters above bars indicate statistically equivalent means. Fill colors of bars are maintained between some figures to facilitate comparisons.

Approximately twice as many symbionts per ascidian were found in Mövenpick reef as in either of the two other sites ($p < 0.001$, one-way ANOVA [log-transformed data]; Figure 3). This pattern was largely related to the significantly higher abundances of the amphipod *L. furina* and the copepod *B. projecta* at that site ($p = 0.023$ and $p < 0.001$, respectively, Kruskal-Wallis; Figure 4). Interestingly, different measurements of symbiont diversity yielded different results. There was a significant difference in species diversity ($p = 0.041$, Kruskal-Wallis; Figure 3), with ascidians from Mövenpick reef having a more diverse symbiont community than those from Zeytouna Beach, and Abu Tig hosts having intermediate and equivalent diversity to the other two populations. In contrast, applying the Simpson Index, a dominance index in essence, did not detect differences among sites ($p = 0.112$, Kruskal-Wallis; Figure 3). As highlighted previously, there were significant differences among sites in the densities of amphipods (*L. furina*) and *B. projecta*, but overall, these two species comprised over 92% of all symbionts found regardless of reef (Figure 4, right panel). The largest number of *L. furina* in a single host was 31 (Mövenpick reef) and for *B. projecta* it was 50, most of which were males (Abu Tig reef).

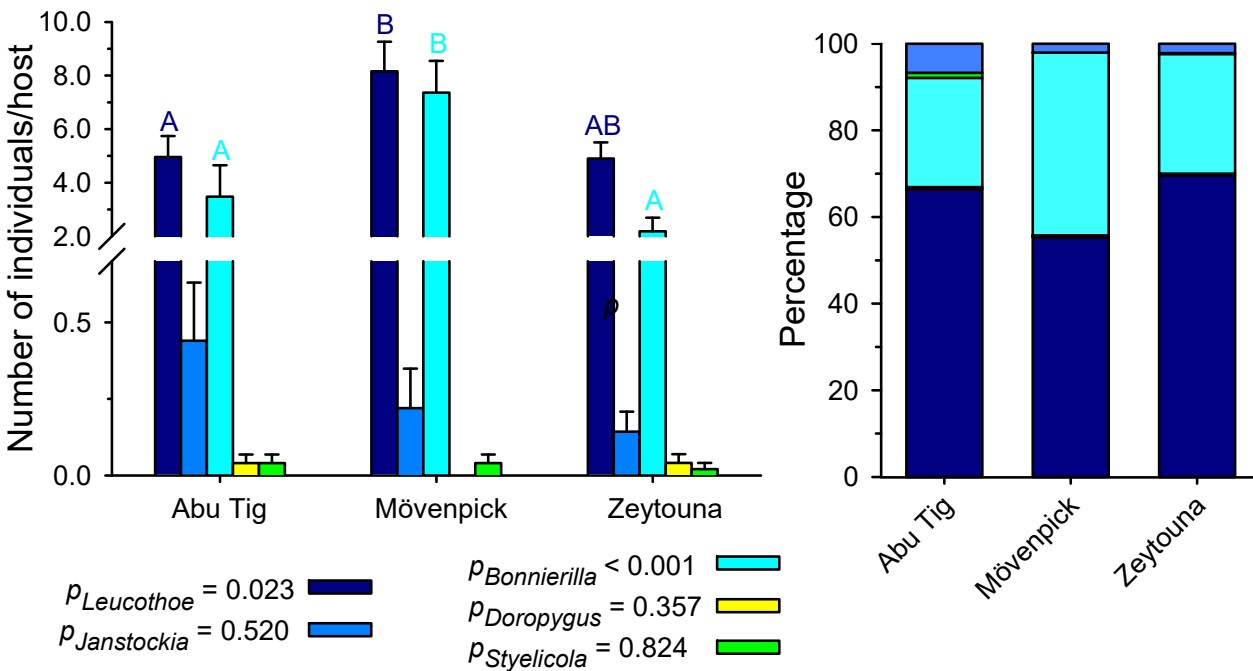

**Figure 4.** Numbers per host and relative abundances of *Phallusia nigra* symbionts collected at three reefs from El Gouna, Egypt. Left graph: Bars represent means ± 1 SE and *p* values are from Kruskal-Wallis tests on individual species. Same letters above bars indicate statistically equivalent means. Right graph: The same data are presented as percentages of total symbiont fauna for reference. These data are not analyzed. Fill colors of bars are as in the left figure.

To evaluate the potential effects of symbionts on their ascidian host, linear regressions were used (Figure 5, Table A1). No significant relations between amphipod, total copepods, or total symbionts were found against host WM, AFDM/WM, tunic AFDM/WM, or body AFDM/WM when data from all three reefs were pooled (Table A1). There was a weak but significant positive relation between total copepods or total symbionts, and host condition index; with a non-significant trend in the same direction for amphipods. When spatial variation was explored by analyzing data from the three reefs separately, few but stronger relations were observed. Data indicated that the total amount of symbionts was positively related with host WM ($p = 0.027$, $R^2 = 0.100$; Figure 5, Table A1) and that *P. nigra* AFDM/WM was also positively related to totals symbiont load (also $p = 0.027$, $R^2 = 0.100$; Figure 5, Table A1), but that these patterns only occurred in Zeytouna Beach. Similarly, *P. nigra* condition index was positively related to amphipod ($p = 0.020$, $R^2 = 0.110$; Table A1), total copepod ($p < 0.001$, $R^2 = 0.212$; Table A1), and total symbionts ($p < 0.001$, $R^2 = 0.363$; Table A1, Figure 5), only at Zeytouna Beach.

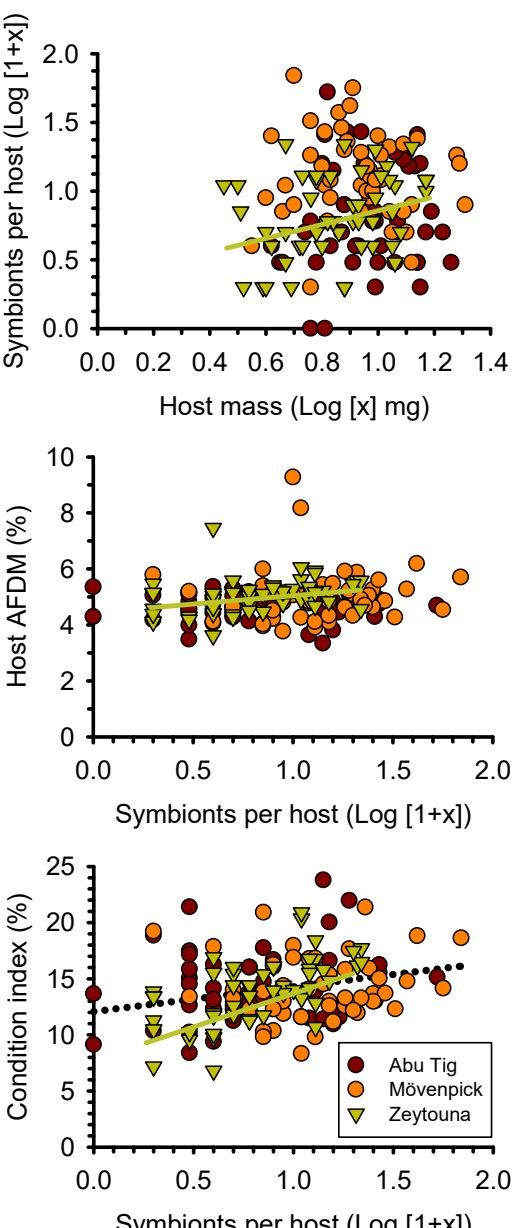

**Figure 5.** Selected regression analyses between host and total symbionts showing the variation of these patterns among reefs. Colored regression lines indicate the reef for which a significant relationship was found. The dotted line on the bottom graph shows the only case in which a general relationship between the two variables considered was found when all data for the three reefs were pooled. See Table A1 for results of all other comparisons performed.

## 4. Discussion

In this study, no strong negative or positive impacts of symbionts on their ascidian host were observed, consistent with a commensalistic interaction. Although a few relationships between host traits and symbiont densities could be detected, the predictability of those patterns was low (Figure 5, Table A1). Analyses also did not detect any evidence of tradeoffs in abundances, competition, or antagonism between different symbionts (Table 2). Inside the ascidian, the location of the symbionts is specific (Figure 2), which could result in decreased competition. For example *S. omphalus* attaches to the visceral mass of *P. nigra*, whereas *J. phallusiella* is found internally on the tunic, and *B. projecta* occurs in the pharyngeal basket [46]. However, the amphipod *L. furina*, and the copepods *B. projecta* and *D. humilis*, all share the pharyngeal basket of the host without apparent exclusion of one

another. In fact, among the very few significant correlations between symbionts, there was a positive (albeit weak) correlation between the abundances of the two most abundant symbionts, *L. furina* and *B. projecta* (pooled data and for Mövenpick reef), and also between *B. projecta* and *D. humilis* at Zeytouna Beach. These results suggest that these symbionts were not resource or space limited in the ascidians studied, and the environmental variables favoring one species would also favor the others. Space limitation may still operate for symbionts that specialize on organs or structures other than the relatively spacious host pharynx. Gage [75] hypothesized that an ascidicolid copepod was found overwhelmingly as single individuals inside their host because they associated with the ascidian food string, whereas a notodelphyid from the same host was found in densities as high as 17 per ascidian within the pharyngeal basket. Interestingly, the ascidicolid *S. omphalus* was always found as single females attached to the visceral mass of *P. nigra* during our study. The lack of negative correlation should not be interpreted as a complete rejection of antagonistic interactions, however. Although there is very little information on the diet of leucothoid amphipods, gut content analysis of a few species suggests those species feed on detritus and crustaceans [76]. If *L. furina* preys on symbiotic copepods living in the ascidian pharynx, a positive correlation can occur as long as the predator is not overexploiting its prey.

Host traits in general did not affect symbiont abundance or diversity. Despite prior studies showing positive correlations between ascidian size and symbiont numbers [77–79], that was not the case here. As seen in Figure 3, *P. nigra* from Abu Tig reef were significantly larger than those from Zeytouna Beach and similar in size to those at Mövenpick reef. However, the number of total symbionts was very similar at Abu Tig and Zeytouna, while Mövenpick reef ascidians contained almost twice as many associated animals. None of these patterns matched the observed anthropogenic influences on these reefs (see Methods). Symbiont diversity (Shannon-Wiener Index) was also significantly higher at Mövenpick than Zeytouna, but no dominance by any given symbiont was observed across reefs (Simpson Index). Thiel [58] found no relation between ascidian mass and numbers of a symbiotic amphipod, whereas Saito [80] found a negative non-linear relation between host mass and density of the copepod *Idomene purpurocincta* [=*Xouthous purpurocinctum* (Norman & Scott T., 1905)]. Both studies used dry mass as proxy for ascidian size, a less accurate approximation of ascidian structure, considering the high water content of the hosts (e.g., about 90% of WM in the *P. nigra* studied here).

Ascidian AFDM/WM showed the opposite pattern to size, with Abu Tig animals having a significantly lower organic content than those from Zeytouna Beach, but regression analyses yielded no indication that these patterns were related to amphipod, copepod, or total symbiont load. More importantly, we hypothesized that condition index of the ascidians could serve as indicator of the relation between symbionts and host: an inverse relation would indicate a negative effect of symbionts on host health (i.e., parasitism), while no relation would be consistent with commensalism, where the symbionts benefit at no expense from the host. Surprisingly, a positive overall relationship was observed between total copepods (considered often as parasites) and host condition, and between total symbionts and host condition when all three reefs were pooled (Figure 5, Table A1). These patterns appeared mostly influenced by the data from Zeytouna Beach, where a much stronger significant positive relationship between densities of amphipods, copepods and total symbionts, and host condition was detected (Figure 5, Table A1). Despite these results, to classify the relationship as a pairwise or diffuse mutualism is not supported. Firstly, only two of the 149 hosts samples were totally free of symbionts, precluding a thorough assessment of host health in the absence of any associated fauna. Second, the comparisons among reefs emphasized the role of spatial variance in understanding patterns. Our data would have suggested different interactions had we sampled only Zeytouna Beach (where a positive correlation was consistent across all symbiont groups and host condition), in comparison to the other two sites. Finally, while useful, condition indices such as the ones calculated here cannot be used as proxies of host fitness without further refinement. In fact, different indices are not equally accurate parallels for animal health and fitness for the

same species, and the same index may not be equally applicable to different species, sexes, or ages [24,25,81–83]. An estimation of gonad mass or reproductive output of *P. nigra* in relation to our calculated body condition index (i.e., gonadosomatic index) would greatly improve the application of this metric on ecological work as it would provide a more appropriate description of fitness (e.g., [14,25]).

The wide range of symbiont loads inside the ascidians sampled here (0–68) also points to a low per capita impact of the symbionts on the host. It is recognized that density-dependent effects are important in changing the nature of symbiotic relations through a parasite–mutualist continuum. Animals providing a net service to a host will fundamentally operate as parasites if their density exceeds a certain threshold [6,8,84,85]. Here, while some symbionts were consistently rare (e.g., *D. humilis*, *S. omphalus*), others varied at least one order of magnitude without any of our analyses detecting negative impacts on the ascidian host.

Our results support the historical treatment of leucothoid amphipods as commensals [45,48,52,53]. For the much more diverse symbiotic copepods [54–56], the existence of both commensal and parasitic species has been recognized [55,86]; yet, the tendency to classify Ascidicolidae and Notodelphyidae as parasitic without further assessment is widespread in the literature [86–90]. In some instances, conclusions about the nature of the interaction were reached after examination of mouth parts, formation of galls or cysts, and position of the symbiont in the host (e.g., [87,91–93]). Those are not unreasonable approximations; the formation of such structures or the intake of host fluids could result in reductions in host performance and fitness. However, feeding on host materials and induction of abnormal tissue growth occurs also with mutualists, such as senita moths, rhizobia, and gall-forming fig wasps [94–98]. A broader analysis of costs and benefits can avoid overgeneralizations about species for which little information, inability for manipulation, or historical treatment of certain related groups, have obscured our understanding of ecological interactions. Other recent studies on invertebrates [13] and vertebrates [12] have highlighted the need to reassess marine symbioses for groups that have been classified as symbionts and parasites.

## 5. Conclusions

The use of condition indices could help elucidate the nature of symbiotic interactions for instances in which symbiont loads cannot be manipulated in the host to quantify performance. Nevertheless, the application of these indices to ecological questions requires further refinement to include more directly related measurements of fitness, such as fecundity or gonad development and mass. A promising additional tool is the measurement of key stable isotopes in host and symbiont to establish the trophic status of the interacting animals [99]. For the Red Sea species studied here, the effects of five different symbionts on the host *P. nigra* appeared minimal, even for copepods with adaptations suggesting that their nutrition comes directly from host fluids or tissues (e.g., *J. phallusiella* and *S. omphalus* [46]) and despite the simultaneous presence of more than one symbiont in a single ascidian. Similarly, symbiont density was not shown to affect hosts within the variance sampled here. Data also suggested that local conditions could influence the trajectory of interactions, as evidenced by some significant patterns observed in single reefs alone. To avoid misclassification of host–symbiont interaction, geographically relevant sampling should be considered.

**Author Contributions:** Conceptualization, E.C.-R.; methodology, E.C.-R.; formal analysis, E.C.-R. and T.L.; investigation, E.C.-R., M.-E.-D.S. and S.E.-S.; resources, E.C.-R.; data curation, E.C.-R.; writing—original draft preparation, E.C.-R.; writing—review and editing, E.C.-R., M.-E.-D.S., S.E.-S. and T.L.; visualization, E.C.-R. and T.L.; project administration, E.C.-R.; funding acquisition, E.C.-R. All authors have read and agreed to the published version of the manuscript.

**Funding:** This research was funded by an American University in Cairo Research Development Grant to the lead author and by funds to M.-E.-D. Sherif from the Department of Biology of the American University in Cairo in support of a senior undergraduate thesis.

**Institutional Review Board Statement:** Not applicable.

**Data Availability Statement:** The data presented in this study are openly available in figshare at https://doi.org/10.6084/m9.figshare.17677787.

**Acknowledgments:** We thank the private administrations of Abu Tig Marina, Mövenpick Hotel, and Zeytouna Beach for access to our field sites. We gratefully acknowledge the staff of the J. D. Gerhart Field Station for logistical support during collections and sample processing.

**Conflicts of Interest:** The authors declare no conflict of interest. The funders had no role in the design of the study; in the collection, analyses, or interpretation of data; in the writing of the manuscript, or in the decision to publish the results.

## Appendix A

**Table A1.** $p$ values from linear regression analyses between symbiont abundances and various measurements of their ascidian hosts (*Phallusia nigra*). Note that host mass was treated as an independent variable against total amphipods (*Leucothoe furina*), total copepods (all four species found together) and log (1 + x) of all symbionts together. However, for ease to accommodate results in this table, host wet mass is placed on the top row. For all other analyses, amphipods, total copepods, and log (1 + x) of all symbionts were treated as the independent variable. Percent ash-free dry mass per wet mass (AFDM/WM) of the ascidian tunic and body were added to obtain total AFDM/WM. Analyses were conducted for all studied reefs together and individually. Numbers in bold indicate significant relations between variables. For those cases, $R^2$ is provided in parentheses. Only positive regressions were detected. Note that $p$ and $R^2$ values were coincidentally similar in two separate analyses.

| All Field Sites | | Host Wet Mass (log x) | Host Total AFDM/WM | Host Tunic AFDM/WM | Host Body AFDM/WM | Host Condition Index |
|---|---|---|---|---|---|---|
| | Amphipods | 0.644 | 0.444 | 0.960 | 0.829 | 0.054 |
| | Copepods | 0.944 | 0.219 | 0.666 | 0.871 | **0.016 (0.039)** |
| | All symbionts (log 1 + x) | 0.088 | 0.137 | 0.976 | 0.928 | **0.001 (0.068)** |
| Individual sites | | | | | | |
| Abu Tig | Amphipods Copepods All symbionts (log 1 + x) | 0.596 0.777 0.359 | 0.639 0.914 0.602 | 0.471 0.838 0.588 | 0.904 0.555 0.128 | 0.635 0.351 0.173 |
| Mövenpick | Amphipods Copepods All symbionts (log 1 + x) | 0.407 0.442 0.756 | 0.854 0.493 0.692 | 0.796 0.317 0.519 | 0.748 0.620 0.799 | 0.282 0.221 0.617 |
| Zeytouna Beach | Amphipods Copepods All symbionts (log 1 + x) | 0.113 0.410 **0.027 (0.100)** | 0.260 0.100 **0.027 (0.100)** | 0.976 0.606 0.653 | 0.989 0.427 0.780 | **0.020 (0.110)** **<0.001 (0.212)** **<0.001 (0.363)** |

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
