# Peer review of "Spatial Variability in a Symbiont-Diverse Marine Host and the Use of Observational Data to Assess Ecological Interactions"

_diversity, doi:10.3390/d14030197_

Round 1

Reviewer 1 Report

Authors have done extremely well on this paper in dealing about symbionts and its ecological interactions. I recommend following things to be added.

  1. Authors need to write about the biology of the host sea squirt.
  2. Table need to be given to show all the symbionts associated with the host. the name of symbionts and its taxonomy.
  3. checklist table if provided, would be nice. The studied host and symbionts in the world.
  4. Pictures need to be provided with the different symbionts and host.
  5. Conclusion is provided with too many citations. Rephrase it, better to start with your own research to conclude and with support from other sources without providing too many citations. 

Author Response

Reviewer 1

1. Authors need to write about the biology of the host sea squirt.

We have added details about P. nigra on page 2 and about its anatomy in the methods (page 4)

2. Table need to be given to show all the symbionts associated with the host. the name of symbionts and its taxonomy.

Table 1 including this information has been added.

3. Checklist table if provided, would be nice. The studied host and symbionts in the world.

We are unclear about this request. The table includes the geographic areas where the symbionts have been studied.

4. Pictures need to be provided with the different symbionts and host.

A new Figure 2 has the pictures of the symbionts, relative size, and location with the host as per requests of both reviewers..

5. Conclusion is provided with too many citations. Rephrase it, better to start with your own research to conclude and with support from other sources without providing too many citations. 

We have eliminated 12 citations from the Discussion. Unfortunately, the request for additional information by both reviewers added several references, resulting in a longer bibliography than previously.

Reviewer 2 Report

Systematic analyses of (putative) "commensal" symbioses are badly needed. This paper represents an initial attempt to carry out such an analysis in Phallusia nigra in the Red Sea. Judging from the nice selection of references (both on symbiosis and on previous work on symbionts of P. nigra, the authors have the right goals in mind. And I am well aware that the gap between the theoretically ideal experimental design and what's actually achievable in real-life conditions can be high and not always surmountable. Nevertheless,  more is needed to make this a really definitive study. I found myself wanting more information in several areas:

  1. more or different measures of "condition" (I was not persuaded that the indicators used by the authors were necessarily of much relevance as "health" or "condition" indicators; the authors were right to note that gonadal mass would have been useful) . I realize that they can't necessarily go back and extract more information from these particular samples (though if they do have other information in their notes, they should use that information here); nevertheless, I found these indices less than fully persuasive. For that reason, it was hard for me to summon much confidence in the meaningfulness of the various statistical analyses that evaluated the relationship between animal condition and symbiont number, density, etc.
  2. It would be really helpful to describe the three reefs more fully. How far apart are they from each other? Is there anything ecologically distinctive about each them, in terms of both physical and biological aspects of the habitat? What were the dates of collection? (Possibly I missed that information, but if it's not there, it definitely needs to be added)
  3. The authors really need to provide a diagram (or perhaps a photo, though I think a diagram might be more useful) that shows the precise anatomical location in the host of each of the symbionts that are being discussed. (Also, FYI, I found "viscera" to be an overly vague term)
  4. Similarly, a simple diagram showing the anatomy and size of each of the symbionts, preferably drawn to scale, to give the reader a sense of the comparative size of each symbiont, would also be useful.
  5. Also, the English still needs some polishing.

Author Response

Reviewer 2

1. (...) more or different measures of "condition" (I was not persuaded that the indicators used by the authors were necessarily of much relevance as "health" or "condition" indicators; the authors were right to note that gonadal mass would have been useful). I realize that they can't necessarily go back and extract more information from these particular samples (though if they do have other information in their notes, they should use that information here); nevertheless, I found these indices less than fully persuasive. For that reason, it was hard for me to summon much confidence in the meaningfulness of the various statistical analyses that evaluated the relationship between animal condition and symbiont number, density, etc.

The reviewer points out twice that he/she is not convinced the application of a condition index is compelling here. In many respects we agree with the reviewer and discuss extensively the limitations of all these approaches (which the reviewer also acknowledges). We also point out that a gonadosomatic index is likely more informative. However, the use of conditions indexes based on body to exoskeleton ratios is a cornerstone of shellfish fisheries and aquaculture. The parallels between meat:shell ratios and body:tunic ratios we used is made more explicitly now in page 5. We would have liked to use a gonadosomatic index, but getting the gonads cleanly dissected was not possible, especially if we aimed to include the male reproductive system.

2. It would be really helpful to describe the three reefs more fully. How far apart are they from each other? Is there anything ecologically distinctive about each them, in terms of both physical and biological aspects of the habitat? What were the dates of collection? (Possibly I missed that information, but if it's not there, it definitely needs to be added)

This information is included in the Methods now. A panel has been added to Fig. 1 also. Some references about the reefs around El Gouna have been added as well. I am aware there are other data but they have been published in predatory journals with dubious publications standards.

3. The authors really need to provide a diagram (or perhaps a photo, though I think a diagram might be more useful) that shows the precise anatomical location in the host of each of the symbionts that are being discussed. (Also, FYI, I found "viscera" to be an overly vague term)

We have added a figure with this information as requested by both reviewers. We did not include it originally because of the variable quality of the only pictures we have, but we have conformed to this request to the best of our abilities.

We also agree with the inadequacy of the terms “viscera” and “visceral mass.” These were derived from the location of these internal organs in a region called the “visceral cavity” and used for convenience to identify symbiont location within the host. The visceral mass is all organs except the pharyngeal basket. We define this now in page 4 and have reduced the use of these terms by referring to the ascidian “body” when the distinction between pharynx and viscera is not required (e.g., Table 2)

4. Similarly, a simple diagram showing the anatomy and size of each of the symbionts, preferably drawn to scale, to give the reader a sense of the comparative size of each symbiont, would also be useful.

We include scale bars in the figure with the symbionts.

5. Also, the English still needs some polishing.

We have included all modifications suggested in the marked manuscript provided by the reviewer

Round 2

Reviewer 1 Report

Yes, it can be accepted in present form.